

# Genomic complexity and clinical significance of the RCCX locus

Vladimir V. Shiryagin[1], Andrey A. Devyatkin[2], Oleg D. Fateev[1], Ekaterina S. Petriaikina[1], Viktor P. Bogdanov[2], Zoia G. Antysheva[2], Pavel Yu Volchkov[2,3], Sergey M. Yudin[1], Mary Woroncow[3] and Veronika I. Skvortsova[4]

[1] Federal State Budgetary Institution "Centre for Strategic Planning and Management of Biomedical Health Risks" of the Federal Medical Biological Agency (Centre for Strategic Planning of FMBA of Russia), Moscow, Russia
[2] Federal Research Center for Innovator and Emerging Biomedical and Pharmaceutical Technologies, Moscow, Russia
[3] Department of Fundamental Medicine, Lomonosov Moscow State University, Moscow, Russia
[4] The Federal Medical Biological Agency (FMBA of Russia), Moscow, Russia

Corresponding authors
Vladimir V. Shiryagin,
VShiriagin@cspfmba.ru
Pavel Yu Volchkov,
volchkov@genlab.llc

## ABSTRACT

Nearly identical, repetitive elements in the genome contribute to the variability in genetic inheritance patterns, particularly in regions like the RCCX locus, where such repeats can lead to structural variations. In addition, during the formation of gametes as a result of meiosis, variants of loci with repetitive elements that do not code for the required proteins may occur. As a result, an individual with certain genetic rearrangements in this region may have an increased risk of developing a congenital disorder, particularly in cases where the non-functional allele is inherited dominantly. At the same time, there is still no routine or generally recognized diagnostic method to determine the sequence of the repetitive fragments. The functionally important RCCX locus consists of such repetitive fragments. The available knowledge about the genomic variants of the RCCX locus is fragmented, as there is no standardized method to determine its structure. It should be noted that in some structural variants of the RCCX locus, the sequence of protein-coding genes is disrupted, leading to the development of diseases such as congenital adrenal hyperplasia (CAH). Although genetic testing is generally accepted as a gold standard for CAH diagnosis, there are a myriad of strategies on which exact methods to use and in which order. The reason for this inconsistency lies in the complexity of the RCCX locus and the fact that each patient or carrier may have a highly individualized mutation or combination thereof. In this review, we have discussed all known methods that can be used to study the structure of the RCCX locus. As a result, optimal approaches are proposed for the diagnosis of the most common disease caused by lesions in the RCCX–CAH due to *CYP21A2* deficiency.

## INTRODUCTION

The first version of the human genome was identified by Sanger sequencing in 2001 (*Lander et al., 2001*). It should be noted that the described sequence of the human genome was not complete—almost identical repetitive regions can be identified by sequencing only in

those loci where the length of the read is greater than the length of the repetitive element of the genome. Recent advances have led to the development of more accessible sequencing methods, as in high-throughput sequencing. As a result, whole genome short sequencing has become a routine practice in clinical laboratories. The human genome, except for the Y chromosome, was not fully sequenced until 2022 when long-read sequencing (*Nurk et al., 2022*) was employed to address the remaining gaps. To date there is no routine and generally accepted common sequencing method available to determine the sequence of repetitive fragments.

The RCCX locus is a region in the human genomic DNA that encompasses several genes: Serine/Threonine Kinase 19 (*STK19*, alias name **RP1**), Complement 4 (**C4**), *CYP21A2*, Tenascin X (*TN**X***). However, the number of copies of this genomic locus may vary. For example, as *Saxena et al. (2009)* report, in their sample population of European ancestry Americans, 60.7% of healthy individuals had four copies of the locus per diploid genome, 26.1% had three and 9.8% had five copies. The presence of multiple copies of homologous genes allows for different variants of homologous recombination during meiosis in sperm and egg formation, leading to the emergence of RCCX locus genetic diversity. Unequal crossing over promotes major structural rearrangements and changes in the number of copies of each locus fragment, which can be considered as one of the main causes of *CYP21A2* gene disruption (*Carrozza et al., 2021*). As a result, the patient develops congenital adrenal dysfunction (congenital adrenal hyperplasia, CAH) due to 21-hydroxylase enzyme deficiency, a product of the *CYP21A2* gene. Furthermore, based on the structure of RCCX, a discrete number of *CYP21A2* haplotypes can be identified that are associated with the development of CAH (*Bánlaki et al., 2013*).

This review has two objectives. Firstly, to describe those common variants of the RCCX locus which cause CAH and are important for our understanding of the disease origins. Secondly, available laboratory diagnostic methods are discussed which can enable the detection of incongruities in the RCCX locus structure. The current review is intended for medical geneticists, endocrinologists and specialists in the field of laboratory genetics.

## SURVEY METHODOLOGY

We searched PubMed (http://www.ncbi.nlm.nih.gov/PubMed) and Google Scholar (http://scholar.google.com) to query the articles cited to select all possible studies with keywords including "congenital adrenal hyperplasia", "*CYP21A2*", "*RCCX*", "*STK19* ", "*C4*" and "**TNXB**" were used to search for relevant articles. The literature was searched for relevant articles from 1986 to 2023. The most appropriate and relevant studies were selected after reading the abstracts.

## STRUCTURE OF THE RCCX LOCUS

### "Canonical" structure of the RCCX locus

The RCCX locus is a multigene cluster located on the 6th human chromosome at position 6p21.1 within the HLA class III region. The particular importance of this locus is related to the location of the 21-hydroxylase P450c21 (*CYP21A2*) gene within it. The *CYP21A2*

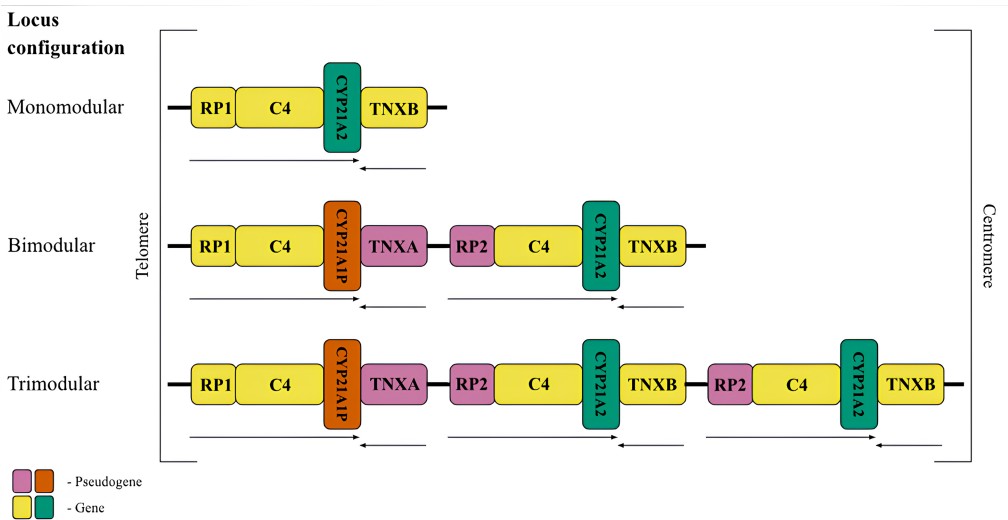

**Figure 1** **The most frequent RCCX copy number variation in the population.** Schematic representation of the possible conûguration of the canonical locus structure. From top to bottom: monomodular configuration; bimodular configuration; trimodular configuration. The CYP21A2 gene was colored green, CYP21A1P was colored red. Other genes were colored gray. The arrows indicate the orientation of the open reading frames (ORFs) of the genes.

gene consists of 10 exons with a total length of 3.4 kilobases (kb) (*Higashi et al., 1986*). Congenital adrenal hyperplasia (CAH) is a group of autosomal recessive diseases with the most prominent feature being hormone synthesis impairment in the adrenal glands. Mutations in the 21-hydroxylase gene, leading to enzyme deficiency, are the most common cause of CAH (*Marino et al., 2022*), making the *CYP21A2* gene the most studied part of RCCX.

The peculiarity of the RCCX locus structure explains the high frequency of mutations within it (*Baumgartner-Parzer, Witsch-Baumgartner & Hoeppner, 2020*). The locus consists of basic homologous units or modules (Fig. 1B) which is the result of a tandem duplication (*Sinnott et al., 1990*). Each module is about 30 kb in length and consists of four genes or their corresponding pseudogenes. The formation of pseudogenes is related to the tandem nature of the duplication. The locus elements are located sequentially on the chromosome (*Sinnott et al., 1992*).

The most common allele in the Caucasian population (69%) (*Blanchong et al., 2000*) consists of two modules of RCCX (Fig. 1, centre), and therefore this configuration deserves detailed consideration. The distal module contains the *STK19* gene (*RP1*, *G11*), which encodes a serine/threonine kinase (*Gomez-Escobar et al., 1998*). The RP1 gene is followed by the *C4* gene, which codes for the 4th component of the complement system in blood plasma. There are long and short forms of the complement gene, which differ from each other depending on the presence of an endogenous retroviral HERV-K insert (*Schneider et al., 2001*). The *C4* protein is one of the most polymorphic components of the complement system and plays a fundamental role in the function of innate immunity (*Wang & Liu, 2021*). Its isotypes C4A and C4B have 99% identical DNA sequences and differ only by

four amino acid residues, but have distinct structures and functions (*Wang & Liu, 2021*). The *C4* gene is followed by the *CYP21A1P* pseudogene. This full-length pseudogene does not correspond to a functional product due to three mutations in the coding sequence, such as an 8 bp deletion in exon 3, a single-nucleotide frameshift in exon 7 and a point mutation in exon 8 (*Higashi et al., 1986*). The first module ends with the *TNXA* pseudogene corresponding to exons 32-44 of the tenascin (*TNXB*) gene. The *TNXA* pseudogene partially overlaps the position of the *CYP21A1P* pseudogene.

The proximal RCCX module contains the paralogs of the listed genes and pseudogenes in the same order. The *RP2* pseudogene, a partial copy of the RP1 gene, precedes the second complement gene *C4*. In most genotypes, the bimodular locus in each of the modules carries an alternative *C4* isotype. This is the only gene that retains full functionality in both modules. The complete absence of the C4B isotype is a risk factor for certain diseases (*Jaatinen et al., 2003*). Another position, closer to the centromere, is occupied by the *CYP21A2* gene (also called *CYP21B* in earlier studies) encoding the enzyme 21-hydroxylase. Due to the particular importance for human metabolism, most RCCX genotyping methods aim to detect variants of this gene. The last fragment is occupied by the functional tenascin gene *TNXB*, which is located on the opposite DNA strand and partially overlaps with the 3′-non-coding sequence of *CYP21A2*. The product of this gene is an extracellular glycoprotein associated with embryonic development (*Burch et al., 1997*) and connective tissue remodeling (*Flück et al., 2008*).

## RCCX locus haplotypes

Less frequent copy number variants in the healthy population include monomodular (17%) and trimodular (14%) RCCX loci (*Blanchong et al., 2000*). Examples of such haplotypes are shown in Fig. 1 (top and bottom, respectively). There is evidence of exceptional cases with either four (*Parajes et al., 2008*) or five modules (*Zhou et al., 2021*). Complement genes within the normal locus may be long or short, depending on the presence of a widespread HERV-K insertion (Fig. 2, top). Despite significant internal homology, the locus exhibits a high degree of polymorphism, leading to the formation of various structural variants, deletions and point mutations.

According to the generally accepted hypothesis, structural variants arise through the mechanism of unequal crossing over (*Sinnott et al., 1990*). The best-known pathogenic structural defect of the monomodular structure is a 30 kb deletion with the formation of a chimeric structure linking partial sequences of the *CYP21* gene and pseudogene (Fig. 2, centre). Another pathogenic structural variant is represented by a large deletion with a complete loss of the *CYP21A2* gene and the formation of a chimeric structure consisting of the pseudogene and the tenascin gene sequences (Fig. 2, bottom).

*CYP21A2*. Another mechanism leading to diversity in the RCCX locus is gene conversion (*Collier, Tassabehji & Strachan, 1993*), which is a significant source of small deletions and point substitutions, particularly in the *CYP21A2* gene. These mutations often result from microconversions (*Strachan, 1994*) between the *CYP21A2* gene and its pseudogene CYP21A1P. In addition to large deletions and gene rearrangements, pseudogene-derived variants, including those caused by gene conversion events, contribute to most pathogenic

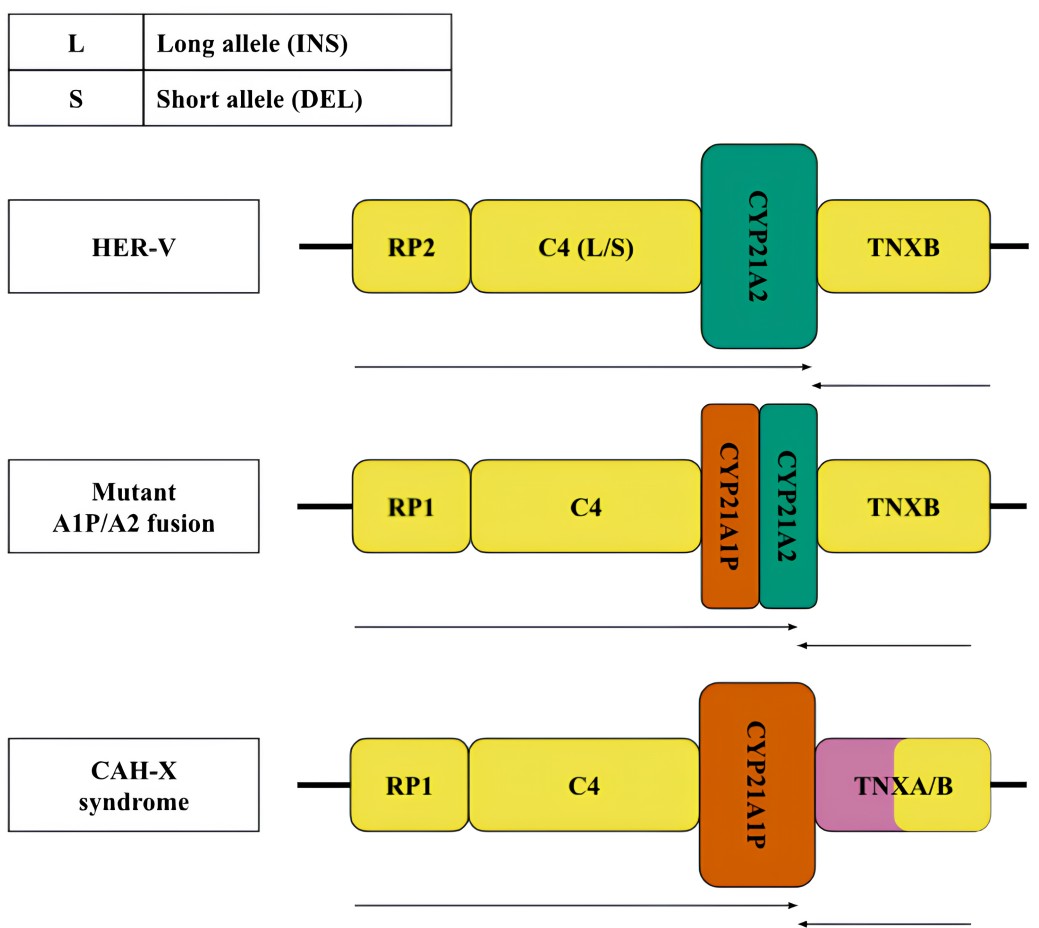

| L | Long allele (INS) |
|---|---|
| S | Short allele (DEL) |

**Figure 2** **Structural variants, large and most frequent deletions and insertions with indication of their clinical significance.** Variants of the RCCX locus: top—normal (HER-V insertion), middle and bottom—pathological—chimeric pseudogene and gene resulting from a 30 kb deletion and CAH-X deletion, respectively. The CYP21A2 gene was colored green, CYP21A1P was colored red. Other genes were colored gray. The arrows indicate the orientation of the open reading frames (ORFs) of the genes.

alleles associated with CAH. The positions of these small deletions and mononucleotide substitutions within the *CYP21A2* sequence and the associated CAH forms are shown in Fig. 3. A more detailed list of pathogenic variants and their descriptions was recently published by *Concolino & Costella (2018)*.

## Diseases associated with RCCX locus haplotypes

The mutual arrangement and similarity of the gene sequences that make up the multimodular RCCX locus contribute to the emergence of phenomena such as structural rearrangements, mutations and variations in the overall length of the locus (*Carrozza et al., 2021*). However, most RCCX haplotypes retain at least one functionally active copy of each locus gene in their structure, which is necessary for the formation of a healthy phenotype. The inability to synthesize a properly functioning protein leads to the development of specific clinical manifestations with a rather clear genotype/phenotype correlation.

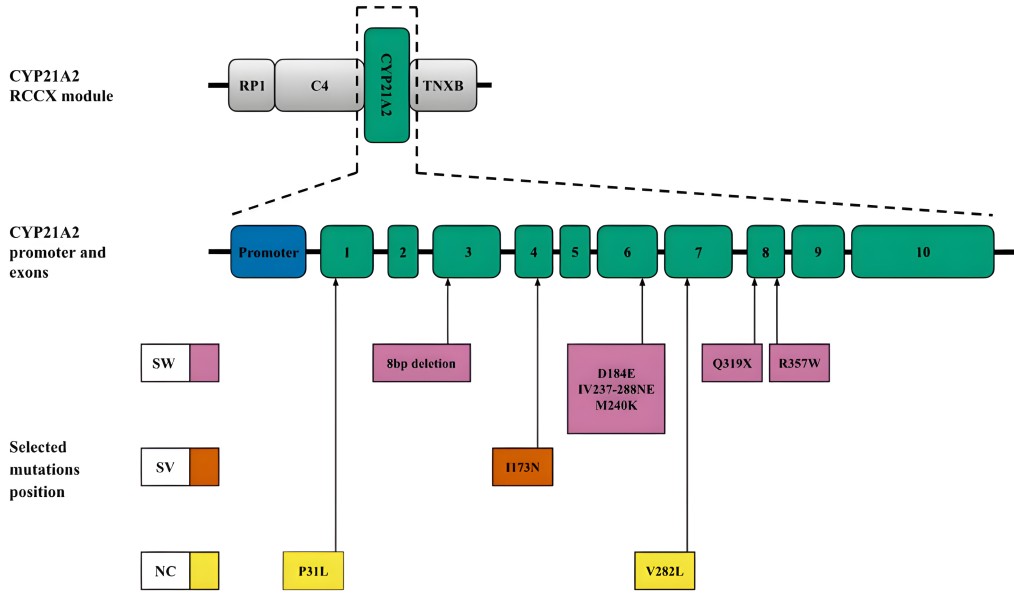

**Figure 3** **CYP21A2 variants with indication of clinical significance.** Top, schematic position of CYP21A2 in the context of the RCCX module. Middle, promoter and exons of the respective gene. Bottom, position of the selected mutations. The colour code corresponds to the CAH form. Abbreviations: SW, salt-wasting form; SV, simple virilising form; NC is a non-classical form.

## RP-dependent disorders

The RP or *STK19* gene, located at the beginning of the locus, encodes the nuclear transcription factor serine threonine kinase 19, which was recently found to be involved in DNA damage response mechanisms (*Boeing et al., 2016*). The significance of this macromolecule with a predominantly nuclear localisation for human health has not yet been fully investigated. The results of genome-wide association studies have shown an association between structural variants of this domain and the development of certain oncological diseases (*Yin et al., 2019*). Particularly in melanoma cells, *STK19* has been shown to activate oncogenic signaling pathways through selective phosphorylation of the *RAS* mutant variant (*NRAS*) (*Gimple & Wang, 2019*). Pharmacological inhibitors of *STK19* in turn blocked *NRAS* phosphorylation and interrupted melanoma cell growth (*Yin et al., 2019*). A number of other researchers, however, consider the connection between the structure of *RP1* and melanogenesis to be unproven (*Rodríguez-Martínez et al., 2020*).

## C4-dependent disorders

*C4* is a gene that codes for one of the complement system proteins. The presence of both major isoforms is necessary to retain normal functioning. In a monomodal haplotype only one of the isoforms is present on the allele, which can lead to a complete absence of either *C4A* or *C4B* in a given individual. Deficiency or complete absence of the *C4A* component is associated with the development of systemic lupus erythematosus and the development of autoimmune diseases in general (*Yih Chen et al., 2016*; *Jüptner et al., 2018*). It is also reported that an isolated increase in the number of copies of the *C4A* gene is a risk factor
for schizophrenia, while an increase in copy number of both isoforms is associated with a higher likelihood of developing Alzheimer's disease (*Zorzetto et al., 2017*; *Yilmaz et al., 2021*).

## CYP-21-related disorders

Congenital adrenal hyperplasia (CAH) is a group of diseases associated with a disturbance in the synthesis of steroid hormones by the adrenal glands. A disease develops due to a dysfunction of one of the enzymes involved in the process of hormone biosynthesis. In about 95% of cases the defect lies within the enzyme 21-hydroxylase, which is encoded by the *CYP21A2* gene at the RCCX locus. This gene is expressed in the adrenal glands during the embryonic development stage (*Voutilainen & Miller, 1986*) starting approximately 50 days postconception (*Claahsen-van der Grinten et al., 2022*) and encodes the 21-hydroxylase—a microsomal enzyme that firstly converts 17-hydroxyprogesterone to the cortisol precursor 11-deoxycortisol and secondly catalyses the reaction progesterone to deoxycorticosterone, the aldosterone precursor (*White et al., 1994*). Thus, the absence of this enzyme results in the inability to produce both glucocorticoid and mineralocorticoid hormones to various degrees. In turn, the synthesis of adrenal androgens does not depend on the activity of 21-hydroxylase. On the contrary, the increased synthesis of ACTH triggered by the absence of glucocorticoids and mineralocorticoids in the blood, results in an excessive production of male sex hormones by the adrenal glands, leading to virilisation of the external genitalia in girls and premature sexual development in boys (*Claahsen-van der Grinten et al., 2022*).

The severity of androgen excess and the mineralo-/glucocorticoid deficiency combined with varying degrees of reduction in 21-hydroxylase function caused by different mutations determines the clinical symptoms of CAH. In complete absence or preservation of no more than 1% of enzyme activity, the most dangerous classical salt wasting (SW) form of the disease develops. At 1–2% of enzyme activity the amount of aldosterone secreted is sufficient to prevent the development of salt wasting crisis and a simple virile (SV) form of the disease develops. A further increase in the proportion of preserved enzyme activity leads to a decrease in the amount of androgen produced and, accordingly, to a lower degree of external virilisation. At 20–50% of the preserved activity, a non-classical (NC) form of the disease occurs, the clinical manifestations of which may not appear until puberty (*Concolino & Costella, 2018*). Thus, to completely prevent the development of the disease, a copy of a healthy gene or a combination of mutant variants with a high level of conserved enzyme activity is sufficient.

Currently, many pathogenic structural variants of *CYP21A2* have been described. A general database of variants can be found at the *Human Cytochrome P450* (*CYP*) Allele Nomenclature Committee (*Pharmacogene Variation Consortium, 2018*) at *Concolino & Costella (2018)* and at *Baumgartner-Parzer, Witsch-Baumgartner & Hoeppner (2020)*. About 95% of mutations leading to *CYP21A2* deficiency are thought to be the result of intergenic recombination at the RCCX locus, which is explained by the relatively close proximity and high homology between the *CYP21A2* gene and the *CYP21A1P* pseudogene. Up to 75% of recombination variants are microconversions of non-functional pseudogene regions into the gene region. The remaining mutations mostly occur by unequal crossing

over and represent deletions affecting the *CYP21A2* region, mostly forming or resulting in CYP21A1P/*CYP21A2* chimeras (*Carrozza et al., 2021*). As with other autosomal recessive diseases, *de novo* events in this gene are rare and are expected to account for 1% of all pathogenic variants (*Concolino & Costella, 2018*). These rarely observed mutations do not arise from the pseudogene and occur naturally.

## CYP21A2CYP21A2CYP21A1P CYP21A2CYP21A2

### *TNXB-associated disorders*

*TNXB* is a gene that codes for one of the connective tissue glycoproteins tenascin-X (TN-X), an important component of fibrogenesis. A deficiency in this protein caused by mutations in the protein-coding fragments of the *TNXB* gene leads to a decrease in fibrinogen density and consequently to a decrease in the elasticity of the connective tissue. Mutations in *TNXB* are the cause of a variant of Ehlers-Danlos syndrome, a disease associated with impaired collagen synthesis and manifested as joint hypermobility, overstretching of the skin and general fragility of the connective tissue.*TNXB*. While *TNXB*-EDS is characterized by autosomal recessive inheritance (*Carrozza et al., 2021*), it is important to note that CAH-X syndrome, which involves hypermobility type Ehlers-Danlos syndrome (EDS) linked to mutations in *TNXB*, follows an autosomal dominant inheritance pattern. This distinction is crucial for understanding the variable expressivity and penetrance associated with *TNXB*-related disorders (*Lao & Merke, 2021*). Heterozygous variants in in *TNXB* have been associated with vesicoureteral reflux, skeletal muscle weakness, primary myopia and heart defects (*Allamand et al., 2014*; *Tokhmafshan et al., 2019*). However, currently the evidence for the gene-disease association is considered as limited for autosomal dominant conditions.

A common variation in gene structure is a 30 Kb deletion of the region between *TNXA* and *TNXB*, resulting in a *TNXA*/*TNXB* chimera, which results in a concomitant deletion of *CYP21A2*. Three structural variants of chimeras have been described in the literature. In the case of a homozygous variant of the deletions under consideration, common CAH and EDS symptoms develop, which are referred to as CAH-X syndrome (*Merke et al., 2013*; *Miller & Merke, 2018*; *Merke & Auchus, 2020*; *Carrozza et al., 2021*). Overall, about 10% of CAH cases are CAH-X syndromes (*Morissette et al., 2015*).

## METHODS OF RCCX LOCUS HAPLOTYPE IDENTIFICATION

To comprehensively characterize the RCCX locus, it is crucial to distinguish between the number of modules and the presence of functional genes and pseudogenes. The RCCX locus is known for its variability in the number of modules, which affects the gene content and the potential presence of functional and non-functional elements. There are a number of methods for this, which are described below. It should be noted that each of the methods has its own advantages and disadvantages, as shown in Fig. 4.

### Laboratory methods of CAH diagnostics in clinical practice

### *Southern blot*

Historically, the oldest method for typing the RCCX locus is the analysis of the length distribution of DNA restriction fragments in peripheral blood lymphocytes. This method

| Method / Comparison | Large deletion/ SV | Minor deletion/ conversion | SNP | CNV | Time | Equipment/ consumables/ qualifications | Source | Comments |
|---|---|---|---|---|---|---|---|---|
| Southern blot + RFLP | +++ | ++ | ++ | ++ | + | +++ | Koppens 1992, Lobato 1998, Chung 2002, Saxena 2009 | Limited sensitivity outside of common restriction sites |
| MLPA | ++ | ++ | + | +++ | ++ | ++ | Concolino 2009 | Low specificity in some cases |
| qPCR | + | + | + | ++ | +++ | ++ | Doleschall 2022 | Detection of particular CNV alleles |
| PCR+ hybridization | ++ | ++ | ++ | ? | ++ | ++ | Al-Obaidi 2016 | Limited to most frequent mutations. CNV detected by qPCR kit |
| PCR + Sanger sequencing | + | +++ | +++ | ++ | +++ | ++ | Tsai 2011, Al-Obaidi 2016 | Larger rearrangements are hard to detect. CNV detection only possible in certain RCCX layouts |
| NGS with short reads | ? | ++ | ++ | ? | ++ | + | Ma 2017, Gangodkar 2021 | Pseudogene elimination requires an enrichment. Large variant detection depends on enrichment strategy. CNV detection depends on an external method. |
| NGS with long reads | ++ | ++ | +++ | ? | ++ | + | Stephens 2021, Liu 2022, Tantirukdham 2022 | Pseudogene elimination requires an enrichment. CNV detection depends on an external method |

**Figure 4  Summary table of methods.** The main methods of RCCX genotyping are presented in tabular form. Methods are rated with respect to their applicability for certain tasks and their resource requirements are presented on a scale from 1 to 3 (1 being the worst, 3 the best). Each rank is marked with a "+" symbol. The column "Sources" contains links to selected works done with the respective method. "?" indicates whether no information on this method was given in the reference source. Large deletions/SV are large losses of genetic material that can affect large sections of DNA, including genes and regulatory elements. These changes are usually larger than one kilobase (kb) and can affect entire genes or even large regions of chromosomes. Minor deletions/conversions are smaller changes in DNA that may involve the loss of one or more nucleotides, as well as conversions where one section of DNA is replaced by another. These changes are usually smaller than one kb and often affect only a few nucleotides.

was primarily used to determine the copy number of the components of the RCCX locus (Fig. 5).

When this method is used to study *CYP21A2* in particular, the products of the functional gene treated with the restriction enzymes TaqI or BglII form fragments of 3.7 kb and 10 kb in length, respectively (Fig. 5). Due to the reverse spectrum of restriction sites in the pseudogene, fragments of 3.2 kb (for TaqI restrictase) and 12 kb (for BglII restrictase) in length are excised from *CYP21A1P* under the same conditions. The fragments are separated by electrophoresis and stained after Southern blotting (*Lobato, Aledo & Meseguer, 1998*). As a result, in the case of the presence of both *CYP21A1P* and *CYP21A2* two fragments would be seen for each sample treated with the aforementioned restrictase. Otherwise, in the case of the presence of chimeric CYP21A1P/*CYP21A2* locus only one fragment would be visualized in the product of restriction analysis by TaqI and BglII. The copy number of locuses is determined by quantitative analysis of the intensity of the visualized bands.

It is important to note, that apart from large rearrangements detectable by restriction-based methods, point mutations and short variants also play an important role in the development of CAH. Detection of these variants require other techniques which are discussed below.

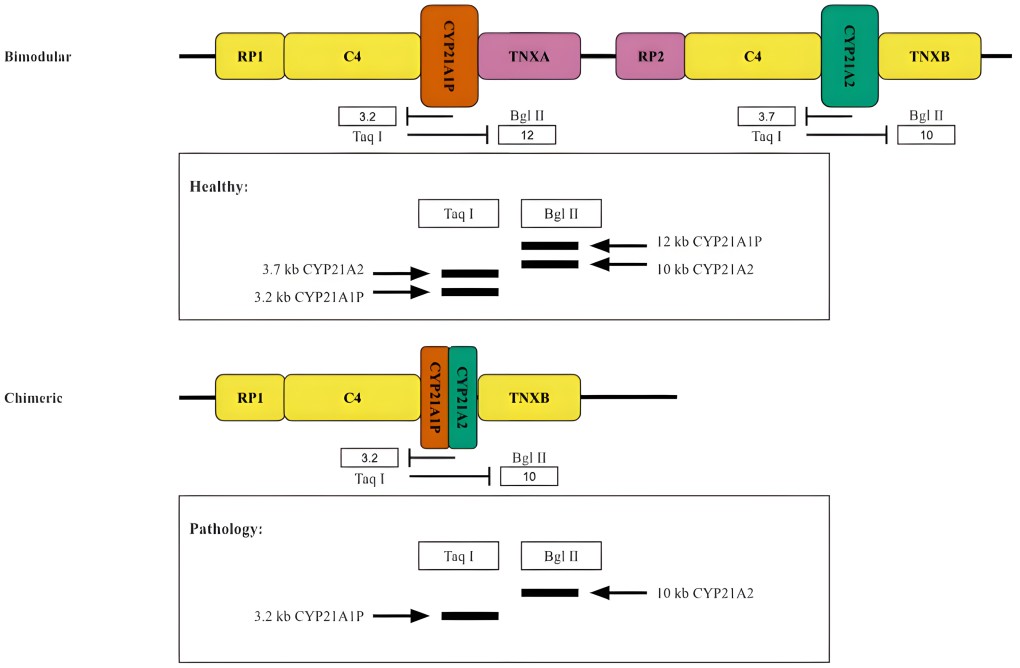

**Figure 5** **Restriction analysis of RCCX locus.** A scheme of two hypothetical cases accompanied by the restriction analysis products. Top, normal case (bimodular locus). Bottom, a common pathology (chimeric locus).

### CYP21A2CYP21A2CYP21A1P
#### *Multiplex-ligation probe amplification (MLPA)*

*CYP21A2CYP21A2.* Multiplex ligation probe amplification (MLPA), also known as MLPA analysis, enables the identification of the copy number variant (*Baumgartner-Parzer, Fischer & Vierhapper, 2007*) as well as the specific point variant of the *CYP21A2* sequence through a multi-step *in vitro* process, followed by computational processing of the results. For this purpose, the genomic DNA interacts with two neighboring complementary probes. Once both probes have hybridized with genomic DNA, they are ligated together. In the next step, the artificial DNA fragment resulting from the ligation of the probe pairs is amplified with specific oligonucleotides labeled with fluorophores. The amplification process leads to an increase in fluorescence, which is measured. It should be noted that MLPA can identify several targets in the multiplex process.

On the one hand, MLPA can detect SNPs, as selected oligonucleotides may or may not hybridize to the region of interest depending on the presence of a SNP. In addition, multiplex MLPA can use a reference region of the genome with a known copy number to compare the result of this reaction with the result of MLPA for the gene of interest (*e.g.*, *CYP21A2*). The difference in copy number between the test fragment and the reference fragment is characterised by the ratio of the peak areas in the electropherogram. The classical application of MLPA is the detection of multicopy variants, but this method is also suitable for the detection of large deletions (*Coeli et al., 2010*). The positive aspects of

the method include high specificity, as PCR template ligation requires precise positioning of hybridised probes on the genomic DNA, and a high degree of standardisation of the protocol. Commercially available reagent kits, optimised by the manufacturer to reduce bias in the detection of a particular locus, provide a reproducible testing technique and facilitate faster training of personnel. The disadvantage of the multi-step and multi-component process is the duration of the procedure (up to several days) and the presence of vendor lock-in.

### Hybridisation of a biotinylated PCR product with enzyme immunoassay detection

Hybridisation detection methods enable researchers to assess the presence of a fixed set of mutation sites within the RCCX locus. For instance, the CAH StripAssay Kit (ViennaLab Diagnostics, Vienna, Austria), based on reverse hybridisation of a biotinylated PCR product with enzyme immunoassay detection, can detect 11 most common point substitutions and approximately 50% of large deletions and conversions known to date (*Al-Obaidi et al., 2016*). While this method does not claim to provide comprehensive coverage of *CYP21A2* variants, it has diagnostic utility in the absence of more sophisticated equipment.

### Array Comparative Genomic Hybridisation (aCGH)

In addition to detecting *CYP21A2* point mutations *via* hybridisation methods, it is also possible to determine its copy number. One of the first genome-wide methods to investigate copy number variation is array comparative genomic hybridisation (aCGH) (*Lucito et al., 2003*). This method in most common two-channel variant is based on comparison between the fluorescence intensity of the genome fragments under investigation (after treatment with restriction enzymes, fluorescent labelling and hybridisation with a library of immobilised oligonucleotides) and a similar indicator for a reference genome containing a known copy number (*De Smith et al., 2007*). The copy number of the RCCX locus in healthy individuals can be determined using gene resolution analysis of copy number variation (graCNV), a method that utilizes an expression microarray platform for aCGH (*Auer et al., 2007*). The advantages of this method include a high degree of standardisation and its widespread use in medical genetics, which provides comparable quality indicators of the results obtained in different laboratories. However, the aCGH method is generally characterised by low sensitivity and requires a significant amount of high-quality genomic DNA. Specific disadvantages include low density of positions detected by the standard probe sets, which limits the resolution within the RCCX locus and does not allow for detection of all common mutations.

### CYP21A2CYP21A2: Quantitative polymerase chain reaction (qPCR)

Real-time PCR or quantitative PCR offers the possibility to detect or quantify target DNA in a reaction mixture. This method is widely used in transcriptomic studies and can be adapted for the study of genomic DNA (*Fernandez-Jimenez et al., 2011*). One such application is the assessment of copy number variations at selected genomic loci (*Weaver et al., 2010*; *Cantsilieris et al., 2014*). In this case, a selected region that is not subject to duplication is used as internal reference.

A commercially available real-time PCR kit (CAH RealFast™ CNV Assay) with dual-channel detection of fluorescently labelled primers (TaqMan) provides semi-quantitative relative copy number determination. The PCR results evaluation consists of two steps. At first, the *CYP21A2* amplification product concentration is calculated relative to the endogenous control. Thereafter, the value obtained is compared to a calibration scale calculated according to samples with a predetermined copy number.

The widespread use of quantitative PCR for RCCX analysis is hampered by a number of obstacles. Problems of a general nature are mainly associated with reaction inhibition due to the complexity of the high molecular weight DNA template (*Sidstedt, Rådström & Hedman, 2020*). In this regard, the initial formulation of the reaction should include the procedure for original sample titration. In addition, the precise selection of the concentration in the sample serves to eliminate the side effect of uneven amplification of the genomic DNA experimental and control fragments. Another difficulty is the tendency for the fluorophore accumulation curve to saturate making it cumbersome to detect copy number at higher values (*Perne et al., 2009*). Nevertheless, in the case of RCCX this factor is of the least importance as copy number in the majority of the population is limited to three modules per allele. Particular challenges in studying the RCCX locus pose the locus' high internal homology and the presence of a large number of variants. Approaches proposed in the literature include parallel analysis of multiple non-overlapping locus fragments or primer multiplexing (*Doleschall et al., 2022*).

## Sanger sequencing
### CYP21A2

Sanger sequencing is routinely used to analyze the *CYP21A2* gene because of its ability to identify small deletions, single nucleotide substitutions and some copy number variations (*Dundar et al., 2019*; *Seroussi, 2021*). The choice of PCR strategy is crucial for preparing the target fragment for sequencing, as it ensures comprehensive coverage of the regions of interest. By complementing long-range PCR and RFLP with Sanger sequencing (*Collier, Tassabehji & Strachan, 1992*; *Stikkelbroeck et al., 2003*; *Coeli et al., 2010*), researchers can assess positions with common variants that may not be detected by the initial assay. This method is particularly useful for the characterization of chimeric breakpoints within the CYP21A1P/*CYP21A2* locus and the identification of single point mutations in functional copies of the *CYP21A2* gene. These capabilities are essential for predicting loss of function and assessing the potential severity of the disease (*Chen et al., 2012*).

## Short-read high-throughput sequencing

The first attempt of studying the RCCX locus *via* short reads has employed a custom target enrichment and second-generation sequencing approach (*Bell et al., 2011*). The authors examined 7,717 regions of 437 genes and reported the discovery of two clinically significant mutations within the *CYP21A2* gene, including mutation CM071683 HGMD (Ala392Thr, 1174G >A). This mutation is of interest because subsequent Sanger studies found at least two false-positive samples carrying an intact *CYP21A2* gene with the true location of the mutation being in the *CYP21A1P* pseudogene. This observation was explained by the fact that the hybridisation probes used to enrich the genomic library with target
regions were equally complementary to both the gene and the pseudogene, so that the amplification products of the gene and the pseudogene could not be distinguished (*Mueller et al., 2013*). As an alternative, the authors proposed using a conventional long-range PCR amplicon which distinguishes functional *CYP21A2* from the pseudogene as a template for subsequent library preparation. In addition, recently the successful implementation of Illumina's DRAGEN™ (Dynamic Read Analysis for GENomics) was demonstrated for the identification of pathogenic *CYP21A2* variants using short-read sequencing data (*Schobers et al., 2024*).

This is the general direction for all subsequent methods based on newer IonTorrent and Illumina platforms for both DNA-based diagnosis including prenatal testing. The *CYP21A2* gene (or CYP21A1 pseudogene) is amplified by specific primers (*Gangodkar et al., 2021*; *Wang et al., 2021*). In addition, a set of multiple target genes can be amplified (*Ravichandran et al., 2021*), but the use of targeted multigene panels counting several hundred genes has proven difficult due to suboptimal RCCX amplification (*Lim et al., 2015*). After amplification the procedure for fragmenting and adding adaptors is similar to that of whole-genome libraries. Most papers describe the formation of paired-end libraries (paired-end reads), but some authors use paired-reads with a large interval (mate-pair reads), which offer extended possibilities for phasing options (*Cradic et al., 2014*). The primary results processing is done taking into account the selective nature of the library preparation step.

Detectable variants for these methods include: deletions of varying length, including a 30 kb deletion leading to formation of a chimeric gene; gene rearrangements; point substitutions. The upside of targeted enrichment approach is its high specificity and the ability to use primers designed in the past for other sequencing methods (*Ma et al., 2014*; *Ma et al., 2017*). Disadvantages include the inability to directly determine copy number during amplification according to the currently proposed schemes. Furthermore, the above studies are mainly based on short read sequencing of the RCCX locus alone, without considering other genome sequences. Thus, a high coverage of the target region is achieved. Nevertheless, the algorithms used in these approaches are of little use to the whole-genome sequencing data already available which in turn hampers population-based studies.

## Emerging directions
### Long read sequencing
Short-read mapping is difficult when sequencing repetitive DNA sequences (*Treangen & Salzberg, 2012*), such as the RCCX locus (*Lee, 2014*). Among the shortcomings of short reads is the lacking option to sequence a full-length locus with multiple RCCX modules. According to researchers, a reliable mapping of reads that allows for a significant difference between the *CYP21A2* gene and its pseudogene is achievable with direct sequencing of an amplicon longer than 5 kb (*Mueller et al., 2013*). Technologies based on long-read sequencing, such as SMRT (PacBio) and ONT (Oxford Nanopore Technologies, Oxford, UK), allow for analysis of contiguous DNA segments at least up to 10–30 kb (*Amarasinghe et al., 2020*; *Dremsek et al., 2021*), which significantly exceeds the stated limit.

Current methods of studying RCCX locus structure with long-read sequencing are based on principles of sequencing target amplicons. Such amplicons can be obtained by locus-specific long-range PCR (*Li et al., 2023*), in particular, from peripheral blood samples (*Tantirukdham et al., 2022*; *Liu et al., 2022*) or dried blood spots (*Liu et al., 2022*). The amplification strategy can be based either on previously developed approaches with a minimal set of locus-specific primers (*Lee et al., 2003*; *Tantirukdham et al., 2022*) or include multiple pairs of primers complementary to both a single gene and the 21-hydroxylase pseudogene. Of note is that possible conversions (*Tantirukdham et al., 2022*) as well as differences in the sequences of the underlying tenascin gene or pseudogene should be taken into account (*Liu et al., 2022*). Based on the amplicons, a library of SMRT bell templates is formed, which are sequenced on the PacBio Sequel II platform with coverage from 20x to 500x. A modification of the method described by Liu et al. gave researchers the opportunity to study up to 5 genes of steroid metabolism in one run by mixing barcoded libraries prepared from amplicons of balanced length. Processing of raw data and interpretation of results were performed according to manufacturer's standard recommendations (*Tantirukdham et al., 2022*). However, some researchers have proposed the use of two specifically prepared reference genomes for a more accurate resolution of A1P/A2 and A2/A1P chimeric reads (*Liu et al., 2022*). The phasing of the haplotypes was carried out using a computational method.

The mutations detected include deletions, duplications, gene rearrangements of varying lengths and single nucleotide substitutions (*Liu et al., 2022*; *Li et al., 2023*). Some difficulties were encountered in interpreting reads assigned to chimeric genes (*Liu et al., 2022*), and method sensitivity depended on the library enrichment strategy using amplicons (*Tantirukdham et al., 2022*). All published experiments were designed exclusively for genotyping RCCX and were not intended to work with genome-wide data.

### Computational improvements

The presence of positions in the human genome that are difficult to map and assemble with next-generation sequencing reads is not limited to the RCCX locus. For example, typing of the pharmacologically important *CYP2D6* gene presents similar challenges, which are overcome by applying specific tools on the computational level (*Chen et al., 2021*).

A particular study (*Ebbert et al., 2019*) was conducted to evaluate various difficult-to-map regions and to explore a computational approach for an analysis with increased sensitivity. In this research, whole genome sequencing was performed on the Illumina platform with reads up to 100 nucleotides long and a mean coverage of at least 33x. The authors detected complex regions, defined as regions with a mapping depth of less than five reads or with a large number of reads with low quality score (90th percentile for a MAPping Quality, MAPQ, range of less than 10) in multiple exons of 748 genes. Of these, 436 genes contained duplications (in some cases multiple) affecting 5% or more of the coding sequence (*Ebbert et al., 2019*). Analysis of such sequences could increase the informational value of the genome-wide data already collected. The authors proposed a two-step strategy for mapping ambiguous reads. The first step is based on sequential masking of duplications in the reference genome and is used to activate reads with ambiguous localisation that are

otherwise assigned a low MAPQ value. The second step is to obtain positional information on the mutation by experimental methods: locus-specific PCR or long-read sequencing in the samples selected in the first step. The authors have demonstrated the effectiveness of their strategy by identifying a mutation in the CR1 gene associated with Alzheimer's disease.

In the context of the expected widespread adoption of long-read sequencing, approaches to improve the analysis and interpretation of this type of sequencing are of particular interest. In these recent or upcoming works, of particular interest are the ones where a method for assessing the differences between pairs of homologous regions is proposed. For example, *Stephens et al. (2021)* proposed a method for finding structural variants based on PacBio long read sequencing used on a mixture of *CYP21A2* and *CYP21A2* P amplicons. The main feature of this approach is the procedure of filtering reads that must contain a sequence of preselected and ordered motifs specific to the 21-hydroxylase gene or pseudogene. Improved filtering of reads allows complex cases of gene rearrangements to be interpreted and copy number to be assessed with an accuracy comparable to that of other experimental methods (*e.g.*, MLPA).

## CONCLUSIONS

The RCCX locus variability is a phenomenon that is responsible for a child developing the CAH phenotype. Even more people are carriers of such mutations. As was shown above, currently applied genotyping methods (Fig. 4) enable investigators to characterise this variability to a significant degree of detailing.

The diversity of CAH diagnostic approaches necessitates the use of a combination of techniques to yield comprehensive results. Classical typing methods include the allele-specific long-range PCR product(which is aimed to filter out the *CYP21A1P* pseudogene sequence) followed by Sanger sequencing combined with restriction analysis or MLPA. The first method is responsible for detecting small deletions and transformations as well as single nucleotide substitutions. The second method determines copy numbers and larger deletions or conversions. However, certain variants and their combinations cannot be detected with methods available, especially when phasing of extended haplotypes is required. Because of the high degree of *CYP21A2* and *CYP21A1P* homology, next-generation short-read sequencing also does not provide a direct answer to those challenges.

An example of a complex case is shown in Fig. 6. The hypothetical carrier of the disease has complex heterozygosity at the RCCX locus. Due to the allele with two intact *CYP21A2* genes, genotyping shows a functional gene in the expected copy number. The allele with two pseudogenes is thus obscured from detection by standard methods.

However, given the numbers of published papers, the most reasonable expectations are associated with the wider acceptance of SMRT long-read sequencing and the advancements in computational algorithms for RCCX locus typing from various sequencing data including currently standard short read whole genome sequencing.

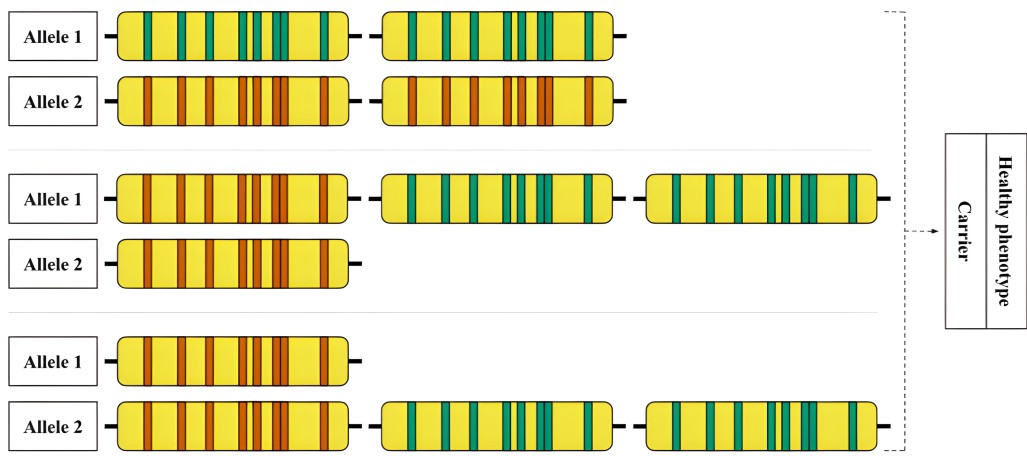

**Figure 6** **An example of an allelic combination of variants that is difficult to interpret.** Hypothetical examples of rare RCCX genotypes that are difficult to detect are shown. Top: Allele 1—locus with two functional CYP21A2 genes, allele 2—locus with two CYP21A1P pseudogenes. Middle: Allele 1—locus with one pseudogene and two functional genes, allele 2—locus with one pseudogene. Bottom: Allele 1—locus with one pseudogene, allele 2—locus with one pseudogene and two functional genes.

# ACKNOWLEDGEMENTS

We sincerely thank Aleksandra I. Akinshina and Dr. Vladimir S. Yudin for their assistance and general support. We thank the biologists of the Centre for Strategic Planning of FMBA of Russia for the valuable and fruitful discussions on the review topic.

## Funding
This study was supported by the Russian Science Foundation (Grant No 23-64-00002). The funders had no role in study design, data collection and analysis, decision to publish, or preparation of the manuscript.

## Grant Disclosures
The following grant information was disclosed by the authors:
Russian Science Foundation: Grant No 23-64-00002.

## Competing Interests
The authors declare there are no competing interests.

## Author Contributions
- Vladimir V. Shiryagin performed the experiments, analyzed the data, prepared figures and/or tables, authored or reviewed drafts of the article, and approved the final draft.
- Andrey A. Devyatkin performed the experiments, analyzed the data, prepared figures and/or tables, authored or reviewed drafts of the article, and approved the final draft.

- Oleg D. Fateev performed the experiments, analyzed the data, prepared figures and/or tables, authored or reviewed drafts of the article, and approved the final draft.
- Ekaterina S. Petriaikina performed the experiments, analyzed the data, prepared figures and/or tables, authored or reviewed drafts of the article, and approved the final draft.
- Viktor P. Bogdanov performed the experiments, analyzed the data, prepared figures and/or tables, authored or reviewed drafts of the article, and approved the final draft.
- Zoia G. Antysheva performed the experiments, analyzed the data, prepared figures and/or tables, authored or reviewed drafts of the article, and approved the final draft.
- Pavel Yu Volchkov conceived and designed the experiments, analyzed the data, authored or reviewed drafts of the article, and approved the final draft.
- Sergey M. Yudin conceived and designed the experiments, analyzed the data, authored or reviewed drafts of the article, and approved the final draft.
- Mary Woroncow conceived and designed the experiments, analyzed the data, authored or reviewed drafts of the article, and approved the final draft.
- Veronika I. Skvortsova conceived and designed the experiments, analyzed the data, authored or reviewed drafts of the article, and approved the final draft.

## Data Availability

This is a literature review.

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
