# Peer review of "Genomic complexity and clinical significance of the RCCX locus"

_PeerJ, doi:10.7717/peerj.18243_

## Round 0.1 · original submission · Major Revisions

The manuscript has been reviewed by three referees. Based on their comments and my reading of it, I find the manuscript requires changes before it can be accepted. I am therefore requesting you to substantially revise the manuscript. Please address all concerns of the referees with changes in the manuscript or statements of rebuttal in the response-to-review letter.

Reviewer 1 ·

Basic reporting

The review article on "Genomic complexity and clinical significance of the RCCX locus" is a good addition to the subject area, addressing a niche lacking substantial literature review. It provides an overview of the RCCX complex and its associated disorders. However, the article requires major revisions to enhance its quality and provide maximum benefit to readers.
Abstract
The abstract should be clearer to ensure that a broader audience, even those with minimal expertise in the field, can gain a clear understanding of the content. Specifically:
Line 29: The statement “Nearly identical, repetitive elements of the genome contribute significantly to the variability of inheritance” needs clarification. The authors should elaborate on what is meant by "variability of inheritance." It is essential to explain how repetitive genomic elements influence inheritance patterns and what specific aspects of inheritance they are referring to.
Line 32: The sentence “As a result, a carrier of such an allele may develop a congenital disorder” does not apply universally to all types of inheritance. The authors should consider that in recessive disorders, carriers typically do not present with phenotypes. This sentence should be modified to accurately reflect the conditions under which a carrier may develop a congenital disorder, possibly specifying whether this applies to dominant or other specific inheritance patterns.
Introduction
Line 54: The sentence “However, the human genome (with the exception of the Y chromosome) was not fully sequenced until 2022 (Nurk et al., 2022), with the additional use of long-read sequencing” should be rephrased for clarity. It should clearly convey that long-read sequencing (LRS) was used to fill in previously unsequenced gaps in the human genome. A suggested revision could be: “The human genome, except for the Y chromosome, was not fully sequenced until 2022 when long-read sequencing (Nurk et al., 2022) was employed to address the remaining gaps.”
TNXB-associated Disorders
Since the authors have mentioned that “TNXB-EDS is a disease with autosomal recessive inheritance,” it is crucial to also discuss the inheritance pattern of CAH-X syndrome, which results in hypermobility type Ehlers-Danlos syndrome (EDS), as autosomal dominant. Reference to a relevant study, such as PMID: 33824469, should be included to support this information.

Experimental design

Review article

Validity of the findings

Review article

Additional comments

None

Reviewer 2 ·

Basic reporting

This review article aims to provide a comprehensive summary of the current understanding of the RCCX region and its clinical relevance to medical geneticists, endocrinologists and specialists in the field of laboratory genetics. Overall, this review article was not well written. It was not well-organized and it is missing the most updated information, especially from the standpoint of a medical or clinical geneticist. In particular, as a review article, literature references should be the ones that can represent the original reports and key contributions from the community. This manuscript didn't do well at this regard. This topic has been recently reviewed (Genes and Pseudogenes: Complexity of the RCCX Locus and Disease. PMID: 34394006). The Introduction adequately introduced the subject and make it clear who the audience is. For one of the most complicated region in our genome, the understanding should be correct (not misleading) and the authors should use the simple language to present.

Experimental design

Article content is within the Aims and Scope of the journal and article type.

Validity of the findings

No comment

Additional comments

I have some comments at the CYP21A2 and TNXB contents as below.

Line 224: “possibly forming” should be replaced by “mostly forming or resulting in”.
Line 225: Should add a sentence to describe that fact that “the remaining rare pathogenic variants (<5% of all pathogenic variants) are not from the pseudogene and occur naturally”. This type is regarded as the third group of the pathogenic variants in CYP21A2, following the first group (gene conversion) and the second group (unequal crossing over).
Line 225: “The rarest mechanism is the appearance of pathogenic structural variants de novo (1%) (Concolino & Costella, 2018).” should be considered to change to “Like other autosomal recessive diseases, do novo events are rare in this gene and are expected to count for 1% of all pathogenic variants (Concolino & Costella, 2018 or other related references).” The authors should understand why de novo variants “seem” to be rare in autosomal recessive diseases. This is not a “rarest mechanism”.

Line 242: CAH-X syndrome was first defined by Dr. Deborah Merke at NIH. Therefore, the references for this should be the original finding (PMID: 23284009) and her review articles (PMID: 32966723; PMID: 29734195).
Line 236-237: “Mutations in TNXB have been associated vesicoureteral reflux…..(Allamand et al., 2014; Tokhmafshan et al., 2019)” should be replaced by “Heterozygous variants in TNXB have been associated vesicoureteral reflux…..(Allamand et al., 2014; Tokhmafshan et al., 2019)”.
Line 237: Should add “However, currently the evidence for the gene-disease association is considered as limited for autosomal dominant conditions.”

Reviewer 3 ·

Basic reporting

The review covers an issue of great interest. The RCCX locus is quite complex and variations can be clinically relevant. There are several tecnique sto study this locus all with advantages and disadvantages. Including the use of NGS tecniques.

I think that the report should clearly distinguish when a technique for characterization of the RCCX locus is intended to focus on determining the number of modules and how many functional gene and pseudogenes are present, or it is intended to identify specific point mutations. This in not always clear especially in the context of the CYP21A2 genes.

Genetics of the CYP21A2 gene should be explained more clearly and maybe in a psecifi parpagraph
a lot of genetics in explaine in the paragraph "RCCX locus haplotipes" and again in CYP21 related disorders.
ex.
- lines 145-151 the are well know pathogenic variants that are pseudogene derived, these account to 95% of pathogenic alleles together with the deletions/ gene conversions. This is not clearly explained.
- lines 216 - 226. A more updated list of variatns is published by Baumgartner-Price et al 2020 (PMID: 32616876).

Experimental design

I think that the report should clearly distinguish when the caracterozation of the lecus is to focus on the RCCX locus in term of number of module and how to determine how many functional gene and pseudogene are present.

In the Methids of RCCS locus haplotype identification it is important to descrive advantages and disadvantages of the tecnique described. more then steps of the tecnique.

line 249: restrictin analysis shoud be replaced by Southern blot. and the tecnique explanation would be benefit of a rewriting. In figure 4 plase indicate in the gene figure where and which enzyme cut the DNA.

MLPA section
The descriptio of MLPA is a confusing, There is 1 kit to study the locus.
The important message here is what kind of resutls can be optained. both information on copy number variant as well specifi point variant are obtained.

line 289 Hybridization based tecnique should besplitted in specifi tecniques (MLPa and Souther blot are also based on hybriization)

line 304 Auer et al describe a special aCGH approach which is not the routine platform. this is not cleary explained and missleading.

Sanger sequecing
The message of this pragraph is quite unclear. Sanger is routinely used for CYP21A2 gene analysis. the PCR strategy to prepare the target fragment to sequence is important and shold be discussed probably. Sanger can e used to characterize a chimeric breakpoint as well to idetify single point mutation in a functional copy of the CYP21A2 gene.

Shortread sequecing
illumina RAVEn module to analyza the CYP21A2 gene should be mentioned.

Validity of the findings

no comment

Additional comments

Gene names should be in italic.

line59. here the previous name of the STK19 (RP1) gene should mentioned as the R in the RCCX come from RP. rpbably it should be choisen as a name to use in the rest of the manuscript to facilitate the reader.

Figure 2: legends. Arrows represent the gene ORF orientation. The colours usega need to be explained again without need to go the figure 1

Figure 3. The mutations (pathogenic variants) in the CYP21A2 gene need to be properly updated and corrected with all the 9 well known pseudogene derived mutations
the mutation S494N is a common polymorfism see ClinVar Variation ID: 193596
SW, SV, Nc are not explained in the manuscript and the mutatin associated phenotype is not relevant in this manuscript.

Fifure 5.
what is the difference between large deletion and mionor deletion/conversion ? clarificatin if needed.
Does "?" mean "not relevant" or "not applicable" or "not possible".

---

## Round 0.2 · accepted · Accept

The revised manuscript satisfactorily addresses all the concerns raised by the three reviewers.